# Effect of Mosses and Long-Term N Addition on δ^13^C and δ^18^O Values of Respired CO_2_ Under a Temperate Forest Floor

**DOI:** 10.3390/plants14172707

**Published:** 2025-08-31

**Authors:** Xingkai Xu, Yuhua Kong, Erpeng Feng, Jin Yue, Weiguo Cheng, Dmitriy Khoroshaev, Sergey Kivalov

**Affiliations:** 1State Key Laboratory of Atmospheric Environment and Extreme Meteorology, Institute of Atmospheric Physics, Chinese Academy of Sciences, Beijing 100029, China; 15236902569@163.com (E.F.); yuejin@mail.iap.ac.cn (J.Y.); 2Department of Atmospheric Chemistry and Environmental Science, College of Earth and Planetary Sciences, University of Chinese Academy of Sciences, Beijing 100049, China; 3College of Forestry, Henan Agricultural University, Zhengzhou 450046, China; y.kong@henau.edu.cn; 4Faculty of Agriculture, Yamagata University, Tsuruoka 997-8555, Japan; cheng@tds1.tr.yamagata-u.ac.jp; 5Institute of Physicochemical and Biological Problems in Soil Science, Russian Academy of Sciences, Pushchino 142290, Russia; d.khoroshaev@pbcras.ru (D.K.); kivalov@pbcras.ru (S.K.)

**Keywords:** δ^13^C, δ^18^O, CO_2_, CH_4_, moss, N addition, driving factors, plant–soil interaction

## Abstract

Static chambers combined with isotopic (δ^13^C and δ^18^O) and flux (CO_2_ and CH_4_) measurements were applied, to explore the effects of mosses and long-term nitrogen (N) addition at two levels (22.5 and 45 kg N ha^−1^ yr^−1^) on δ^13^C and δ^18^O values of respired CO_2_ across three autumn seasons under a temperate forest (northeastern China) and their relationships with CO_2_ and CH_4_ fluxes and with soil properties. Mosses generally depleted δ^13^C and enriched δ^18^O in respired CO_2_, likely by altering soil microenvironments or/and substrate use. The effect of N addition on the δ^13^C and δ^18^O values of respired CO_2_ varied with years, and its interaction with mosses had no effects on the isotopic values. The removal of mosses decreased CO_2_ fluxes and the addition of N at a high dose increased CH_4_ fluxes. The δ^13^C and δ^18^O values of respired CO_2_ decreased at soil moisture levels below and above an optimum, and the moisture-dependent effect became more pronounced for the δ^18^O than for the δ^13^C. The results of structural equation modeling showed that 70% of the variability of δ^13^C values of respired CO_2._ was accounted for by the N addition, mosses, soil moisture, and CH_4_ and CO_2_ fluxes, while only 22% of the variability of δ^18^O values of respired CO_2_ was explained by these factors. The results highlight that moss–soil interaction drives the isotopic shifts, which is modulated by N availability. Soil moisture regulates the δ^18^O values of respired CO_2_, but its drivers remain poorly understood. Future work should target processes influencing the δ^18^O shifts of respired CO_2_ and deep soil property interactions.

## 1. Introduction

Within terrestrial ecosystems, respiration and photosynthesis have contrasting impacts on δ^13^C and δ^18^O values of atmospheric carbon dioxide (CO_2_), via the discrimination against ^13^CO_2_ and C^18^O^16^O [1,2,3]. In the last three decades, many previous studies have documented that climate (precipitation, temperature, and light), vegetation types, and soil properties can significantly affect atmospheric CO_2_ concentration within terrestrial ecosystems and the δ^13^C and δ^18^O values of ecosystem-respired CO_2_ on time scales [1,4,5,6,7,8,9,10]. Unfortunately, due to a significant scarcity of site-specific δ^13^C and δ^18^O measurements for soil-respired CO_2_ across diverse terrestrial ecosystems [8,9,11,12,13,14], the specific driving factors controlling both δ^13^C and δ^18^O values of CO_2_ released from soil respiration, especially beneath moss-dominated forest floors, remain poorly understood [3,11,15,16,17,18,19].

As a species that prefers shade and dampness, bryophytes are widely distributed under forest floors in different climate zones around the world. It has been shown that the carbon (C) photosynthetically fixed by the forest floor moss species depends on different tree canopies and accounts for approximately 5–50% of gross primary productivity by tree canopy [20,21], probably affecting the soil C cycle and the isotopic discrimination of respired CO_2_. Unlike vascular plants on parallel evolutionary lines, mosses are characterized by the lack of stomata in moss photosynthetic tissues, thus resulting in a major limitation to CO_2_ diffusion in mosses at water content levels above an optimum [22,23]. There was a parabolic relationship between the C assimilation rates of mosses and moss water contents under laboratory conditions, with a maximum rate at an optimum moisture [15]. The conductance to water in mosses would increase with water content below an optimum and then decreased above the optimum moisture. Hence, the response of mosses to ambient water content would have important impacts on the discrimination against ^13^CO_2_ and C^18^O^16^O [15], affecting the changes in the δ^13^C and δ^18^O values of respired CO_2_ on time scales. There was a significant and negative correlation between δ^13^C values of CO_2_ released from forest ecosystem respiration and soil respiration and soil moisture contents at the depth of 40 cm in the Pacific Northwest of North America [24]. Pataki et al. reported that the average δ^13^C values of forest and grassland ecosystem respiration across North and South America had a parabolic relationship with the mean annual precipitation, with a minimum at approximately 2100 mm [4]. Ometto et al. showed that the δ^18^O values of respired CO_2_ in forest and pasture ecosystems were generally more depleted during the wet than during the dry seasons over the year [6]. These studies mostly focused on the changes in the δ^13^C and δ^18^O values of CO_2_ released from forest ecosystem respiration under varying moisture conditions, along with less attention under moss-dominated forest floors. Due to the lack of stomata in mosses, the responses of moss C assimilation and moss respiration to varying moisture levels are different from those in vascular plants, indicating that the responses of the carbon and oxygen isotopic discrimination to increased water availability would be in contrast to those observed in vascular plants [23]. More importantly, the presence of mosses has an important effect on soil nutrients and soil organic matter turnover in forest ecosystems via its biological N fixation and the leakage of moss photosynthates [21,25,26,27,28], which can in turn affect the discrimination against ^13^CO_2_ and C^18^O^16^O during respiration in the moss–soil continuum [11,15,16]. Unfortunately, to date there has been limited information about the responses of δ^13^C and δ^18^O signatures of respired CO_2_ in the presence and absence of mosses to soil moisture contents under well-drained forest floors [11,16]. In this study, we want to explore whether δ^13^C and δ^18^O values of CO_2_ released from the soil–moss continuum under well-drained forest floors would have a maximum at an optimum moisture level, like the moss C assimilation rates versus moss water contents relationship.

With the increase in anthropogenic activities and nitrogen (N) fertilizer usage, the global atmospheric N deposition has increased to 105 Tg N yr^−1^ [29,30], which will inevitably lead to the intensification of atmospheric N input into forest ecosystems. This will, in turn, profoundly affect the key processes of forest underground C and N cycles, partly via the function of mosses. The enhanced atmospheric N deposition and/or the addition of extraneous N, especially at high doses for a long-term duration, can have negative impacts on the characteristics of mosses such as biological N fixation [31,32] and the growth of mosses [33,34,35,36,37,38] in terrestrial ecosystems, which will, in turn, affect the discrimination against ^13^CO_2_ and C^18^O^16^O during photosynthesis and respiration. Following the surplus N input, the leakage of moss photosynthates may decrease with decreasing C assimilation, probably affecting soil C availability and the moss-mediated microbial decomposition of soil organic C. Based on a coupling method for a flow bioreactor and stable carbon isotope analyzer, Min et al. [17] pointed out that carbon availability can affect microbial respiration and microbial C use efficiency, which in turn affects the discrimination against ^13^CO_2_ and C^18^O^16^O during microbial C assimilation and microbial respiration. Furthermore, the decrease in soil pH [39] and easily decomposable components of soil organic C in forest ecosystems [40] following the long-term increased N input would lead to shifts in the δ^13^C and δ^18^O values of CO_2_ released from microbial respiration. Additionally, atmospheric methane (CH_4_) uptake under forest floors would be limited upon N addition [41,42], so that CO_2_ derived from CH_4_ oxidation can partly contribute to the changes in the δ^13^C values of CO_2_ accumulated during the opaque chamber closure period due to natural biogenic CH_4_ sources with an average δ^13^C value of approximately –63.0‰ [43]. Hence, the negative impacts of increased N input on the growth of mosses [33,34,35,36,37,38] and its N-fixing capacity [31,32], along with the following changes in soil properties (e.g., decreased C availability and low soil pH) and CH_4_ uptake [39,40,41,44], would have impacts on the δ^13^C and δ^18^O values of CO_2_ released from the soil–moss continuum under forest floors. Here, we propose that the presence of mosses, changes in soil properties, and CO_2_ and CH_4_ fluxes under the context of long-term N input can affect the changes in the δ^13^C and δ^18^O values of CO_2_ released beneath forest floors.

The objectives of this study are to explore the effect of mosses and N addition on the δ^13^C and δ^18^O values of CO_2_ released from temperate forest floors, using a long-term N manipulation experimental platform, and the main driving factors by considering CO_2_ and CH_4_ fluxes and soil properties. The results will help improve the understanding of how the moss–soil interaction drives the isotopic shifts of CO_2_ released from well-drained forest floors under the context of increased atmospheric N deposition.

## 2. Materials and Methods

### 2.1. Description of Study Site

The experiment was conducted on a gentle slope portion in a secondary mixed deciduous forest near the National Research Station of Changbai Mountain Forest Ecosystems (42°24′ N and 128°6′ E), at the foot of the Changbai mountains, northeastern China. The dominant trees of this forest are white birch (*Betula platyphylla* Suk.) and mountain poplar (*Populus davidiana* Dode), mostly >70 years old, and its stand density is 1402 stem ha^−1^, with a tree canopy coverage of approximately 66% [45]. Under the secondary forest stand, mosses are widely distributed around the base of tree trunks and beneath a forest floor mostly covered with a small amount of litter. The dominant species of mosses in the forest understory bottom layer include *Haplocladium microphyllum* (Hedw.) Broth., *Heterophyllium affine* Fleisch., and *Pylaisia polyantha* (Hedw.) Schimp. The annual mean temperature is approximately 4.1 °C, and annual mean rainfall is approximately 770 mm in the recent two decades, with more than 80% rainfall from May to October. The temperate forest soil is an andosol, with a <2 cm aboveground litter layer and approximately 10 cm depth of A layer. Soil total organic C and total N contents were 56.8 mg C g^−1^ and 4.5 mg N g^−1^, respectively. The bulk density of surface soil at 10 cm depth was approximately 0.96 g cm^−3^. The soil texture is that of silty loamy soil according to USDA-NRCS soil texture classes. The contents of sandy, silt, and clay were 22.9%, 75.9%, and 1.2%, respectively.

### 2.2. Layout of Field Experiment

This study was carried out in the long-term N simulation experimental measurement platform in the forest stand, which was started in October 2010 to measure soil greenhouse gas (GHG) fluxes. The long-term N simulation experiment included the addition of ammonium chloride at low and high doses (22.5 and 45 kg N ha yr^−1^, namely, low N and high N, respectively) and a control without artificial N addition. The low amount of added N corresponded to twice the dissolved total N input via natural throughfall under the forest stand on an annual scale. Within the long-term N simulation experiment, the effects of litter removal and root removal via trenching and their combinations with N addition on soil GHG fluxes were measured annually. Each experimental treatment had four independent plots with an area of 2 m × 8 m, and all plots were randomly interspersed, and separated mostly by more than 15 m wide buffer strips. The amount of added N was equally sprayed on the N-treated experimental plots monthly during the growing season (from May to October) each year, and an equal amount of water (10 L) was added to the control plots. Due to the removal of litter and grasses each year, mosses were distributed on the ground in the litter-free experimental plots. To study whether and how the presence of mosses in combination with the long-term N addition has impacts on the δ^13^C and δ^18^O signatures of respired CO_2_, following 9 years of N additions, measurements of δ^13^C and δ^18^O values of respired CO_2_ were performed in the autumn seasons of 2019, 2021, and 2024 in the litter-free experimental plots using soil collars with and without mosses.

### 2.3. Measurements of δ^13^C and δ^18^O Values of Respired CO_2_ as Well as CO_2_ and CH_4_ Fluxes

In each litter-free experimental plot, two paired UPVC (unplasticized polyvinyl chloride) collars (20 cm in diameter and 20 cm in height) were inserted into the soil at approximately 7 cm depth and placed permanently in the field with and without moss cover, for measurements of δ^13^C and δ^18^O values of respired CO_2_ as well as CO_2_ and CH_4_ fluxes. There were eight paired UPVC collars in total in each treatment. In the autumn seasons of 2019, 2021, and 2024, static chamber equipment was once used to collect gas samples using air-tight 100 mL syringes and 300 mL air-tight gas bags (NDEV41-0.3, Dalian Delin Gas Packing Co., Ltd., Dalian, China) at regular closure intervals (e.g., 0, 5, and 10 min, depending on the intensity of CO_2_ flux) in each experimental plot [46]. Within one week after gas sampling, a CO_2_ isotope spectrum analyzer (CCIA-38d-EP, Los Research Inc., Fremont, CA, USA) with a manual injection model was applied to determine CO_2_ concentration and its ^13^C/^12^C and ^18^O/^16^O isotopic ratios in gas samples (Figure 1). CO_2_ concentration and its ^13^C/^12^C and ^18^O/^16^O isotopic ratios were averaged over a period of 228 s at a frequency of 1 Hz, and the ratios were based on Pee Dee Belemnite (PDB) as the standard, expressed by δ^13^C and δ^18^O:δ^13^C or δ^18^O (‰) = (*R*_sample_/*R*_standard_ − 1) × 100
where *R*_standard_ and *R*_sample_ indicate the molar ratios of standard ^13^C/^12^C (or ^18^O/^16^O) and sample, respectively.

Prior to each assay, the CO_2_ isotope analyzer was calibrated using two standard CO_2_ mixture gases containing two different CO_2_ concentrations (CO_2_ concentrations and its δ^13^C and δ^18^O values: 398.60 µL L^−1^ and −22.31‰ and −31.00‰; 794.30 µL L^−1^ and −22.34‰ and –31.20‰), 1% Argon (*w*/*w*), and balance gas of air. The standard CO_2_ mixture gases were supplied by Air Products (Beijing Helium Pu North Branch Gas Industry Ltd., Beijing, China), and their δ^13^C and δ^18^O values were measured with a mass spectrometer combined with a Trace Gas System (Isoprime-100, Elementar Ltd., Langenselbold, Germany), using reference materials NBS 19 and IAEA-CO-8 by the Environmental Stable Isotope Laboratory, Chinese Academy of Agricultural Sciences. The analytical precision for CO_2_ concentration and its δ^13^C and δ^18^O values using the CO_2_ isotope spectrum analyzer (CCIA-38d-EP, Los Research Inc., Fremont, CA, USA) was 0.05 µmol CO_2_ mol^−1^, 0.1‰, and 1.0‰ for a 5-min integration time, respectively. The δ^13^C and δ^18^O values of respired CO_2_ across all experimental plots were calculated based on the Keeling plot approach [47,48] (Figure 1), namely, using the intercept of linear regressions of the δ^13^C or δ^18^O values of CO_2_ against the reciprocal of headspace CO_2_ concentrations at closure time scales (*n* = 3). The determination coefficient of linear regressions (*R*^2^) was mostly more than 0.97 for the δ^13^C-CO_2_ and more than 0.85 for the δ^18^O-CO_2_. Following the measurement of δ^13^C and δ^18^O values of respired CO_2_, soil samples at 10 cm depth in the N-treated and non-treated experimental plots with mosses were collected to determine soil properties such as bulk density, pH, microbial biomass C and N, dissolved organic C, and dissolved total N.

To effectively compare the relationships between CH_4_ and CO_2_ fluxes versus the δ^13^C and δ^18^O values of respired CO_2_ across N-treated and non-treated experimental plots with and without moss cover, CH_4_ and CO_2_ fluxes in all experimental plots were determined prior to the measurement of δ^13^C and δ^18^O values of respired CO_2_ (Figure 1), using a portable greenhouse gas analyzer (915-0011, Los Research Inc., Fremont, CA, USA) coupled to an opaque smart respiratory chamber (SC-11, Beijing LICA United Technology Limited, Beijing, China) [49]. The sampling frequency of the gas analyzer was 1 Hz for the measurement of CO_2_ and CH_4_ concentrations. During the 3-min closure, the record for the initial 90 s was automatically discarded due to a dynamic balance inside the analysis system, and the following records were used to calculate CO_2_ and CH_4_ fluxes using the slope of linear regressions of headspace gas concentrations against the opaque chamber closure time, along with the surface area of soil collars and atmospheric pressure data recorded on the gas analyzer upon each flux measurement. CO_2_ and CH_4_ fluxes were expressed as μmol CO_2_ m^−2^ s^−1^ and nmol CH_4_ m^−2^ s^−1^, respectively. Soil temperature and volumetric water content (%, *v*/*v*) at 7 cm depth in soil collars were determined using a temperature and moisture combination probe nicely attached to the portable gas analyzer (Figure 1). To study whether there are significant variations in the δ^13^C and δ^18^O values of respired CO_2_ after the removal of moss blankets, CO_2_ and CH_4_ fluxes as well as the δ^13^C and δ^18^O values of respired CO_2_ were measured prior to and immediately after the removal of mosses on 31 October 2024.

### 2.4. Measurements of Soil Properties

Soil moisture content was measured gravimetrically by drying soil samples at 105 °C for 48 h. Soil bulk density at 10 cm depth was determined using the intact soil core method [50]. The pH values of fresh soil (soil/water, 1/2.5, *m*/*m*) were measured using a portable pH meter (PB-10, Sartorius, Göttingen, Germany). Soil microbial biomass C (MBC) and N (MBN) concentrations were determined using the chloroform fumigation and extraction method [51]. Concentrations of K_2_SO_4_-extractable dissolved organic C (DOC) and dissolved total N (DTN) in fumigated and non-fumigated soils were determined by using a TOC/TN analyzer (Shimadzu TOC-V_CSH_/TN, Tokyo, Japan), and MBC and MBN concentrations were calculated by the differences of K_2_SO_4_-extractable DOC and DTN contents between fumigated and non-fumigated soils divided by 0.45 [52,53]. The specific UV absorbance of soil K_2_SO_4_ extracts at 254 nm (SUVA_254_) (L mg^−1^ cm^−1^) was calculated by the absorbance of soil extracts at 254 nm divided by the concentration of DOC and multiplied by 100 [54]. The UV absorbance of soil extracts at 254 nm was measured by using a spectrophotometer (Unic 2800A, Shanghai, China) with a 1 cm path-length quartz cell.

### 2.5. Calculation and Statistical Analysis

All measured variables were examined for normality using a Shapiro–Wilk test and homogeneity of variance using Levene’s test, and log-transformed where necessary. A two-factor repeated measures analysis of variance (ANOVA) with treatment and moss as fixed factors and year as a random factor was used to assess their effects on the δ^13^C and δ^18^O values of respired CO_2_. A univariate repeated measures ANOVA with treatment as a fixed factor and year as a random factor was used to assess their effects on soil properties. Box plots of daily CO_2_ and CH_4_ fluxes from N-treated and non-treated plots with and without mosses across the three autumn seasons were drawn using OriginPro 2021 (OriginLab Corporation, Northampton, MA, USA). A two-factor repeated measures ANOVA with N level and moss as fixed factors was performed to assess their effects on the daily CO_2_ and CH_4_ fluxes in the three autumn seasons, and the δ^13^C and δ^18^O values of respired CO_2_ each year. Pearson correlation analysis was carried out to evaluate the relationships between δ^13^C and δ^18^O values of respired CO_2_ versus CO_2_ and CH_4_ fluxes, and soil properties. All the data were analyzed by SPSS (version 19.0, IBM Corp., New York, NY, USA). Scatter plots were drawn to present the effects of CH_4_ and CO_2_ fluxes and soil moisture on the δ^13^C and δ^18^O values of respired CO_2_, and their relationships were fitted with curve regressions, using OriginPro 2021 (OriginLab Corporation, Northampton, MA, USA). A random forest analysis was performed to assess the variables significantly influencing the δ^13^C and δ^18^O values of respired CO_2_ using the ‘rfPermute’ package. The random forest analysis was performed using R 4.4.1 [55]. The evaluation of model performance was determined using mean squared error (MSE) and the determination coefficients of regressions (*R*^2^). Based on the results of principal analysis, N addition (added N versus control), moss (with moss versus without moss), CO_2_ and CH_4_ fluxes, and soil moisture were selected as predictors to establish a priori structural equation modeling (SEM) to evaluate the direct and indirect effect pathways on the δ^13^C and δ^18^O values of respired CO_2_ under the experimental conditions. The overall goodness of fit for the model was determined using the chi-squared test, comparative fit index, and the root-mean-square error of approximation index. The SEM analysis was performed using the software AMOS 24 (SPSS Inc., Chicago, IL, USA).

## 3. Results

### 3.1. Changes in Soil CO_2_ and CH_4_ Fluxes as Well as δ^13^C and δ^18^O Values of Respired CO_2_ Prior to and After the Removal of Moss Blankets

Immediately after the removal of moss blankets on 31 October 2024, the average soil CO_2_ and CH_4_ fluxes in the control decreased from 0.449 to 0.373 μmol CO_2_ m^−2^ s^−1^ and increased from –0.605 to –0.526 nmol CH_4_ m^−2^ s^−1^, respectively (Appendix A). This indicated that, on average, the presence of mosses contributed to approximately 17% and 13% of the soil CO_2_ and CH_4_ fluxes under the moss-covered forest floors at a given duration. Furthermore, based on paired *t*-test results, the difference in the soil CH_4_ flux caused by the removal of moss blankets became significant at *p* < 0.05 (Appendix A). Prior to and after the removal of moss blankets in the control, there were no significant differences in the δ^13^C and δ^18^O values of respired CO_2_, respectively (Appendix A). This strongly indicated that autotrophic respiration of the moss species mentioned in this study would have no significant impacts on the δ^13^C and δ^18^O values of respired CO_2_ under the well-drained forest floors.

### 3.2. Effect of Long-Term N Addition on Soil Properties Under Moss-Covered Forest Floors

Following the decadal application of NH_4_Cl at low and high doses, soil pH values in all N-treated experimental plots were decreased by 0.4–0.7 units, relative to those in the control (*p* < 0.01) (Table 1). The soil dissolved total N contents and the SUVA_254_ values of soil K_2_SO_4_ extracts were significantly increased with increasing doses of added N (*p* < 0.05) (Table 1). There was a significant interannual difference in soil dissolved total N, pH, and MBC contents (*p* < 0.05) (Table 1). Soil bulk density was significantly affected by treatments and years of soil sampling (*p* < 0.05) (Table 1).

### 3.3. Effects of Mosses and Long-Term N Addition on the δ^13^C and δ^18^O Values of Respired CO_2_ as Well as CO_2_ and CH_4_ Fluxes

Compared with those recorded with the removal of mosses, average δ^13^C values of respired CO_2_ in the control and the low- and high-N treatments with mosses appeared to reduce by approximately 1.03‰, 1.06‰, and 0.60‰, respectively (Table 2). However, there were not significant differences across all treatments (Table 2). Regardless of moss species, the effect of N levels on the δ^13^C and δ^18^O values of respired CO_2_ varied with years (Table 2), with a significant effect for the δ^13^C in 2021 and for the δ^18^O in 2019 and 2024 (Appendix A). Based on the results of ANOVA analysis, the δ^13^C and δ^18^O values of respired CO_2_ were significantly affected by the presence of mosses over the experimental period (Table 2). There were no significant interaction effects between moss and N level on the δ^13^C and δ^18^O values of respired CO_2_ under the experimental conditions (Table 2, Appendix A). Based on the *in situ* measurement of CO_2_ and CH_4_ fluxes across the three autumn seasons, regardless of N levels, there were significant larger CO_2_ fluxes in the presence of mosses, compared with those in the absence of mosses (*p* < 0.01) (Figure 2a). Furthermore, the largest CH_4_ fluxes were observed in the high-N-treated experimental plots with and without mosses relative to those in the control and low-N treatment (*p* < 0.001) (Figure 2b).

### 3.4. Factors Affecting the δ^13^C and δ^18^O Values of Respired CO_2_ Under Well-Drained Forest Floors

The correlation analysis showed that the δ^13^C and δ^18^O values of respired CO_2_ under the experimental conditions were significantly correlated with soil CO_2_ and CH_4_ fluxes (*p* < 0.01), soil temperature and moisture at 7 cm depth (*p* < 0.01), as well as soil DTN content and bulk density (*p* < 0.05) in all moss-covered experimental plots (Table 3). The δ^13^C values of respired CO_2_ across all moss-covered plots had a significantly negative correlation with soil pH (*p* < 0.05) and MBC (*p* < 0.01) (Table 3).

The CO_2_ flux could account for 53% of the variability in the δ^13^C values of respired CO_2_ across all experimental plots (*p* < 0.001), and 9% of the variability in the δ^18^O values of respired CO_2_ (*p* < 0.05) (Figure 3a,d). There was a parabolic relationship of δ^13^C and δ^18^O values of respired CO_2_ against CH_4_ fluxes under the experimental conditions (*p* < 0.01) (Figure 3b,e). The δ^13^C and δ^18^O values of respired CO_2_ under the experimental conditions decreased at soil moisture below and above an optimum, respectively, and the moisture-dependent effect became more pronounced for the δ^18^O than for the δ^13^C (Figure 3c,f).

Random forest analysis showed that soil temperature and moisture, CO_2_ flux, and bulk density were the strongest predictors of δ^13^C values of respired CO_2_ (*p* < 0.05) (Figure 4a), while the δ^18^O values of respired CO_2_ were significantly predicted by the soil moisture and temperature, daily CH_4_ flux, DTN, and MBC (*p* < 0.05) (Figure 4b). The SEM results indicated that 70% of the variability in the δ^13^C values of respired CO_2._ was explained by the N addition, moss, soil moisture, and CH_4_ and CO_2_ fluxes (Figure 5a). The increases in the δ^13^C values of respired CO_2_ were positively associated with the decreases in soil moisture (*p* < 0.001) and with the increases in CH_4_ and CO_2_ fluxes (*p* < 0.05) (Figure 5b). Both N addition and moss would have negative impacts on the δ^13^C values of respired CO_2_ (Figure 5b). A total of 22% of the variability in the δ^18^O values of respired CO_2_ was explained by the N addition, moss, soil moisture, as well as CO_2_ and CH_4_ fluxes (Figure 6a), with a maximum negative effect of soil moisture (*p* < 0.01) (Figure 6b).

## 4. Discussion

### 4.1. In Situ Measurements of δ^13^C and δ^18^O Values of Respired CO_2_

Under the experimental conditions, the selection of chamber closure time (e.g., less than 10 min) in autumn could support the difference in CO_2_ concentrations among headspace gas samples collected during each measurement period being mostly larger than 75 ppm, while the closure time being as short as possible would minimize the disturbance of the soil-chamber system upon each gas sampling [46]. This should be considered an effective method to reduce the standard error in the *y* intercept obtained via the Keeling plot approach [4,46]. Across the three autumn seasons, there were generally nice determination coefficients of linear regressions, *R*^2^, for the δ^13^C and δ^18^O values of respired CO_2_ (mostly 0.97 and 0.85, respectively). Perhaps due to the difference in the measurement precision of δ^13^C and δ^18^O values of CO_2_ using the CO_2_ isotope spectrum analyzer (CCIA-38d-EP, Los Research Inc., Fremont, CA, USA) and the presence of non-steady conditions inside the soil-chamber system [46], *R*^2^ values for the δ^18^O values were mostly smaller relative to those for the δ^13^C values. Nevertheless, as shown in Figure 1, the measurement of δ^13^C and δ^18^O values of respired CO_2_ under forest floors can be nicely performed following a reasonable gas sampling procedure.

### 4.2. Effect of Mosses and Long-Term N Addition on the δ^13^C Values of Respired CO_2_

The δ^13^C values of respired CO_2_ with and without moss cover during the experimental period varied by as much as 6.4‰ and 6.6‰, respectively (Appendix A), indicating an obvious interannual difference in the δ^13^C. A similar change in the δ^13^C values of CO_2_ released from forest ecosystem respiration (4.4‰) and soil respiration (6.2‰) over the growing season was reported by Fessenden and Ehleringer [24] in the Pacific Northwest of North America. The more negative δ^13^C values of CO_2_ respired from all experimental plots with mosses than without mosses (Table 2) indicate that CO_2_ released from moss-induced respiration is more depleted in ^13^C. Due to the discrimination against ^13^CO_2_ during moss photosynthesis, moss biomass normally has a lower ^13^C/^12^C ratio than atmospheric CO_2_ and the following moss-induced respiration will lead to the release of CO_2_ depleted in ^13^C into the atmosphere [3]. At a high dose of added N, the growth of mosses was inhibited [33,34,35,36,37,38], and the difference in daily CO_2_ fluxes in the high-N treatment with and without mosses was, on average, the smallest across all treatments during the three autumn seasons (Figure 2a). This indicated that there would be a relatively smaller discrimination against ^13^CO_2_ during moss photosynthesis and the moss-induced respiration in the high-N-treated experimental plots than in the control, which in turn contributed to a small decrease in the moss-induced average δ^13^C value of respired CO_2_ in the high-N treatment (0.6‰) (Table 2). However, the δ^13^C values of respired CO_2_ in the control had no significant variations prior to and immediately after the removal of moss blankets at a given time (Appendix A), indicating that autotrophic respiration of moss plant would have a small contribution to the δ^13^C values. Thus, changes in the moss-induced soil properties under the experimental conditions would have impacts on the δ^13^C values of respired CO_2_ beneath the moss-species-dominated temperate forest floors (Table 3 and Figure 4a).

The results of SEM (Figure 5) indicated that soil moisture and CH_4_ and CO_2_ fluxes should be considered as major factors controlling the changes in the δ^13^C values of respired CO_2_ under the moss-species-dominated forest floors after a long-term N application. Usually, atmospheric CH_4_ (biogenic source, with a mean of approximately –63.0‰) [43] appears more depleted in ^13^C than normal atmospheric CO_2_ (approximately –8.0‰), and an increased CH_4_ uptake caused by the presence of mosses in the control plots (Figure 2b) can, in turn, lead to more negative δ^13^C values of CO_2_ accumulated during the opaque chamber closure time. Although forest soil CH_4_ uptake in the region, even at low soil moisture, was three orders of magnitude lower than CO_2_ emission [56], the increase in the moss-induced atmospheric CH_4_ uptake could partly contribute to the larger decrease in the moss-induced average δ^13^C values of respired CO_2_ in the control (1.0‰) than in the high-N treatment (0.6‰) (Table 2). A parabolic relationship between soil CH_4_ flux and δ^13^C values of respired CO_2_ across all experimental plots (*R*^2^ = 0.30, *p* < 0.001) (Figure 3b) indicated an important function of CH_4_ uptake in regulating the δ^13^C values of respired CO_2_ under the moss-species-dominated forest floors. Furthermore, a relatively large uncertainty in the δ^13^C values of respired CO_2_ in response to low CH_4_ uptake under the experimental conditions (Figure 3b) showed that other factors would contribute to the δ^13^C values.

In addition to the contribution of CH_4_ uptake, the variations in δ^13^C values of CO_2_ released beneath the moss-species-dominated forest floors would be associated with the relative magnitude of moss cover, forest disturbance, and stand age, as well as heterotrophic soil respiration [3,16,17]. In autumn, the mineralization of deep soil organic matter enriched in ^13^C would be increased due to the legacy effect of summer warming and the input of root exudates, leading to the release of CO_2_ enriched in ^13^C from the moss–soil continuum under boreal forest floors [16]. The SUVA_254_ values of dissolved soil organic matter can, to some extent, reflect the biodegradation of soil organic C under forest floors [54], and the largest SUVA_254_ value of soil extracts recorded in the high-N-treated experimental plots (Table 1) would reflect the relatively low soil C availability. Based on the microbial nutrient stoichiometry theory [57,58], the low soil C availability in the high-N-treated experimental plots would stimulate the activity of soil microorganisms like *K*-strategist to release C-requiring enzymes under N-surplus conditions to degrade the recalcitrant organic C in soil, which is normally enriched in ^13^C. This can, to some extent, support the relatively small decrease in the moss-induced average δ^13^C values of respired CO_2_ in the high-N treatment (0.6‰) (Table 2). Based on the difference in the δ^13^C values between microbial biomass and microbial respired CO_2_, Min et al. [17] reported that carbon availability could promote the microbial respiratory fractionation. This would be consistent with the negative relationship between δ^13^C values of respired CO_2_ and MBC contents under the experimental conditions (*p* < 0.01) (Table 3). Hence, soil available C can, to some extent, affect the variations in the δ^13^C values of CO_2_ released from the moss-dominated forest floors. The negative relationship between the δ^13^C values of respired CO_2_ and soil moisture (Table 3) was in agreement with the results reported by Fessenden and Ehleringer [24], who pointed out that δ^13^C values of CO_2_ released from forest ecosystem respiration and soil respiration were negatively and significantly correlated with soil moisture levels at the depth of 40 cm. Together with the results obtained from the random forest analysis (Figure 4), soil moisture, available C pool, and CO_2_ and CH_4_ fluxes should be taken into account as priority variables to predict the changes in daily δ^13^C values of CO_2_ released under temperate forest floors. The results would have many potential applications in the climate–terrestrial C models for rationally predicting δ^13^C values of respired CO_2_ and soil organic C turnover beneath forest floors at large scales and their responses to the presence of mosses and increasing atmospheric N deposition.

### 4.3. Effect of Mosses and Long-Term N Addition on the δ^18^O Values of Respired CO_2_

The average δ^18^O values of CO_2_ released across all treatments beneath the moss-species-dominated temperate forest floors in northeastern China (Table 2) were close to the average δ^18^O value (–14.4‰) of CO_2_ released during vegetation and soil respiration under Canadian boreal forest floors during the growing season [11]. A similar average δ^18^O value (–13‰) of ecosystem-respired CO_2_ was also observed in Amazonian forests over the year [6]. The difference in the δ^18^O values of respired CO_2_ across site-specific forest stands would be ascribed to different precipitation sources with varying ^18^O signatures [59] and the oxygen isotopic discrimination during photosynthesis, respiration, and transpiration of water in forest ecosystems [6,11,18,60,61].

The δ^18^O values of CO_2_ released under the experimental conditions were mainly influenced by an equilibrium isotope effect that occurs between oxygen in soil water and plant chloroplast water and oxygen in CO_2_ dissolved into the water [3,11,18,62]. As shown in Figure 1, an opaque static chamber was used to collect gas samples at intervals for measuring the δ^18^O value of headspace CO_2_; thus, plant (including mosses and tree roots) and soil respiration as well as evaporation mainly contribute to the changes in the δ^18^O values of CO_2_ released under the experimental conditions. Under high soil moisture, evaporation of water from the moss–soil continuum would increase the discrimination against ^18^O [62], leading to a more negative δ^18^O value of any CO_2_ diffusing out of the moss and soil surface. This would explain the relatively lower δ^18^O values of CO_2_ released across all treatments under high soil moisture (Figure 3f). A much better parabolic relationship between soil moisture and δ^18^O values of plot-respired CO_2_ in the presence of mosses (the determination coefficient of regression, *R*^2^ = 0.50) than in the absence of mosses (*R*^2^ = 0.24) (Figure 3f) would, in turn, support the discrimination against C^18^O^16^O via the evaporation of moss water. Furthermore, the relatively larger uptake of hydrophilic substances in mosses under hydration than under desiccation conditions [63] and a pulse release of readily soluble C from mosses into the soil upon hydration [26] would stimulate the activity of the enzyme carbonic anhydrase in the moss–soil continuum, which may support a greater depletion of respired CO_2_ in ^18^O under high soil moisture [18] (Figure 3f). The more negative δ^18^O values of CO_2_ released under high soil moisture were in agreement with the results reported by Ometto et al. [6], who showed that CO_2_ released from Amazonian forest and pasture ecosystem respiration was generally more depleted in ^18^O during the wet than during dry seasons over the year. Based on the nice parabolic relationship between soil moisture and δ^18^O values of CO_2_ released across all treatments, there would be a maximum δ^18^O value of CO_2_ released at approximately 32.7% soil moisture (*v*/*v*). This indicated that there would be an optimum moisture for the largest δ^18^O values of respired CO_2_ beneath the moss-species-dominated temperate forest floors. At soil moisture below an optimum, soil respiration and plant respiration (including moss respiration and tree root respiration) became more sensitive to drought, and δ^18^O values of respired CO_2_ would decrease mainly due to increased conductance to CO_2_ diffusion out of the soil–moss continuum under dry conditions and the relatively lower ^18^O content in soil water than in plant leaf water [3,18,60].

The optimum soil moisture content recording the largest δ^18^O value of respired CO_2_ under the experimental conditions became larger than that for its maximal δ^13^C value (14.8% soil moisture, *v*/*v*) (Figure 3c,f). Under low moisture conditions, the decline in conductance to water vapor in mosses would increase the conductance to CO_2_ diffusion [15]. The increased CO_2_ diffusion out of mosses and evaporation in the soil–moss continuum would increase the discrimination against ^13^CO_2_ and C^18^O^16^O, thus leading to more negative δ^13^C and δ^18^O values of respired CO_2_ relative to those at an optimum moisture. At an optimum water content, there would be a maximum rate of C assimilation by the mosses [15], which results in CO_2_ depleted in ^13^C and ^18^O entering into plant chloroplast due to discrimination and can, in turn, enhance the release of depleted ^13^CO_2_ and C^18^O^16^O into the atmosphere via respiration [64,65]. However, the high evaporation potential at the optimum moisture level [15] would inhibit the conductance to CO_2_ diffusion in the soil–moss continuum, which can in turn eliminate the release of isotopically depleted CO_2_ into the atmosphere. This trade-off effect of C assimilation and evaporation potential in the soil–moss continuum on the isotopic discrimination against ^13^CO_2_ and C^18^O^16^O would provide some evidence for explaining the largest δ^18^C and δ^18^O values of CO_2_ released at an optimum soil moisture. Under high moisture conditions, the rate of C assimilation by the mosses and the ratio of chloroplastic CO_2_ to ambient CO_2_ concentration would decrease because of the creation of an aqueous diffusion barrier to CO_2_ in mosses, and the evaporation rate remained high [15,22]. Furthermore, the following low temperature under high moisture conditions in autumn would eliminate the growth of mosses, due to the temperature-dependent growth response of mosses [66]. These would lead to less discrimination against ^13^CO_2_ and C^18^O^16^O through moss photosynthesis as water content was increased above an optimum [15,23]. This cannot support the release of depleted ^13^CO_2_ and C^18^O^16^O into the atmosphere under high soil moisture in this study. However, beneath the moss-species-dominated forest floors, the roots of trees were included in all experimental plots, and unlike the lack of stomata in mosses, both ^13^C and ^18^O discriminations in vascular plants would increase with soil water availability due to changes in nonstructural carbohydrate content and carbonic anhydrase activity in the soil–moss continuum, when soil moisture was not saturated [18,67,68,69], thus promoting the release of depleted ^13^CO_2_ and C^18^O^16^O into the atmosphere, mainly via autotrophic respiration of tree roots. More importantly, moss and soil water under high moisture conditions would become depleted in ^18^O due to the new input of precipitation in autumn, relative to moss and soil water under dry conditions [11]. Based on the measurement and simulation, the mean precipitation δ^18^O values in Northeast China became more negative in winter than in summer, resulting from less contribution of warm western Pacific pool (59% in summer versus 36% in winter) with higher δ^18^O values relative to the continental air mass [59]. The equilibrium isotope effect that occurs between oxygen in CO_2_ dissolved in the water and oxygen in soil water and moss/tree roots water [6,11,18,60] would support a higher depletion of respired CO_2_ in ^18^O under high moisture conditions (Figure 3f). The more negative δ^18^O values of CO_2_ released under high moisture conditions was in agreement with the results reported by Ometto et al. [6], who showed that there were more negative δ^18^O values of ecosystem-respired CO_2_ in the Amazonian zone during the wet than during the dry seasons over the year.

## 5. Conclusions

Mosses act as a critical biological filter under forest floors, significantly depleting δ^13^C and enriching δ^18^O in respired CO_2_ under the experimental conditions. This study indicated that moss–soil interactions (e.g., moisture retention, C input, and microbial habitat) are key drivers of isotopic shifts (δ^13^C depletion and δ^18^O enrichment) in respired CO_2_ under the litter-free temperate forest floors. The magnitude and direction of these isotopic effects are modulated by N availability, with a significant cross-year difference. Soil moisture influences the two isotopic shifts, particularly the δ^18^O values of respired CO_2_. While N addition, moss, moisture, and CO_2_ and CH_4_ fluxes explained most of the δ^13^C variations, the drivers of the δ^18^O shifts remain largely unexplained by these factors. Besides these factors, other factors such as the δ^18^O values of plant water and soil water as well as the source of precipitation would need to be taken into account for reasonably explaining the changes in the δ^18^O signatures of respired CO_2_ under site-specific conditions. The stark contrast in the explained variance (70% for δ^13^C vs. 22% for δ^18^O) underscores a greater complexity and the influence of unmeasured processes (diffusion, microbial processes, or atmospheric exchange) on the oxygen isotope signatures of soil-respired CO_2_ compared to its carbon signatures. Future research must prioritize identifying and quantifying these drivers for the δ^18^O shifts of soil-respired CO_2_.

This study provides new insights into the factors driving the δ^18^O and δ^13^C values of CO_2_ released from total soil respiration under a temperate forest floor, yet offers no characterization of the two isotopic signatures for soil respiration components. To advance understanding of the forest ecosystem C cycle under global change, future research would move beyond total soil respiration isotopes and implement targeted, long-term field studies that explicitly measure and differentiate the δ^18^O and δ^13^C signatures of CO_2_ released from soil autotrophic and heterotrophic respiration across various site-specific terrestrial ecosystems. This component-specific data should be crucial for accurate partitioning and modeling of soil C dynamics in terrestrial ecosystems.

## Figures and Tables

**Figure 1 plants-14-02707-f001:**
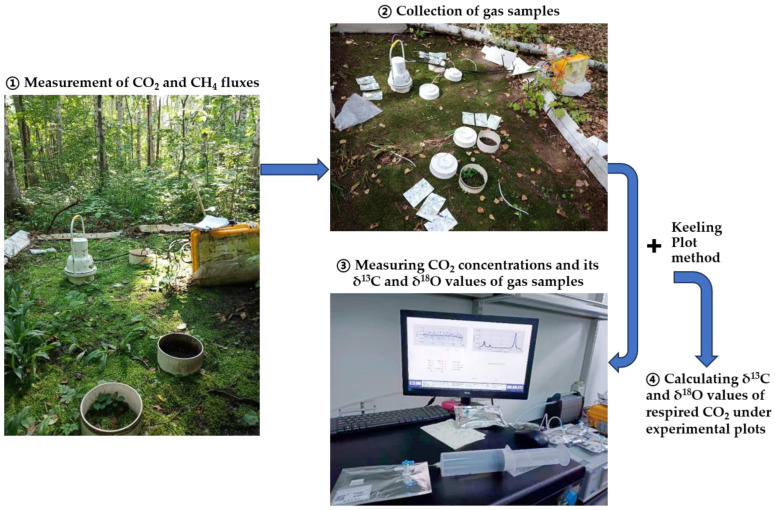
Apparatus for measuring CO_2_ and CH_4_ fluxes, δ^13^C and δ^18^O values of respired CO_2_ under experimental plots. Upon each flux measurement, grasses and litter in all the experimental plots and inside soil collars were removed artificially.

**Figure 2 plants-14-02707-f002:**
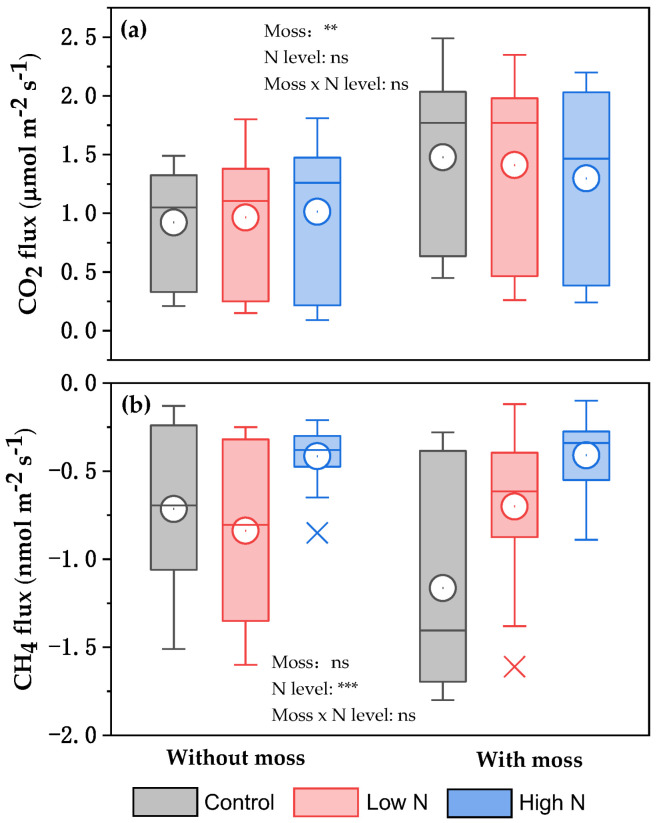
Box plots of daily CO_2_ (**a**) and CH_4_ (**b**) fluxes from N-treated and non-treated experimental plots with and without moss cover in autumn seasons of 2019, 2021, and 2024. Boxes show interquartile (IQR), and circles and horizontal lines in boxes show mean and median values, respectively. Lower and upper whiskers (x) represent 75 percentiles plus 1.5 IQR and 25 percentiles minus 1.5 IQR, respectively. Results of ANOVA are shown in Figure 2. **, *p* < 0.01; ***, *p* < 0.001; ns, not significant.

**Figure 3 plants-14-02707-f003:**
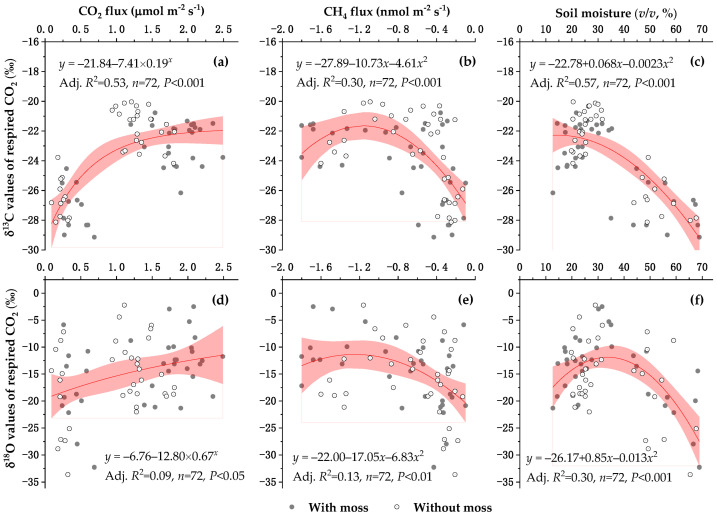
Relationships between δ^13^C (**a**–**c**) and δ^18^O (**d**–**f**) values of respired CO_2_ versus CO_2_ and CH_4_ fluxes and soil moisture at 7 cm depth. Non-linear regressions were fitted according to all data with and without mosses, and 95% confidence bands were shown in red.

**Figure 4 plants-14-02707-f004:**
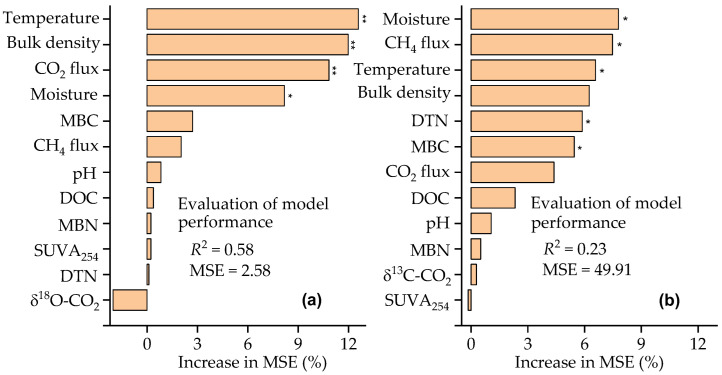
Random forest analysis explaining the effects of environmental variables on the δ^13^C (**a**) and δ^18^O values (**b**) of respired CO_2_. MSE, mean squared error; MBC, microbial biomass C; MBN, microbial biomass N; DOC, dissolved organic C; DTN; dissolved total N; δ^13^C-CO_2_ and δ^18^O-CO_2_ represent the δ^13^C and δ^18^O values of respired CO_2_, respectively. CH_4_ and CO_2_ fluxes represent CH_4_ and CO_2_ fluxes from experimental plots with mosses. *, *p* < 0.05; **, *p* < 0.01.

**Figure 5 plants-14-02707-f005:**
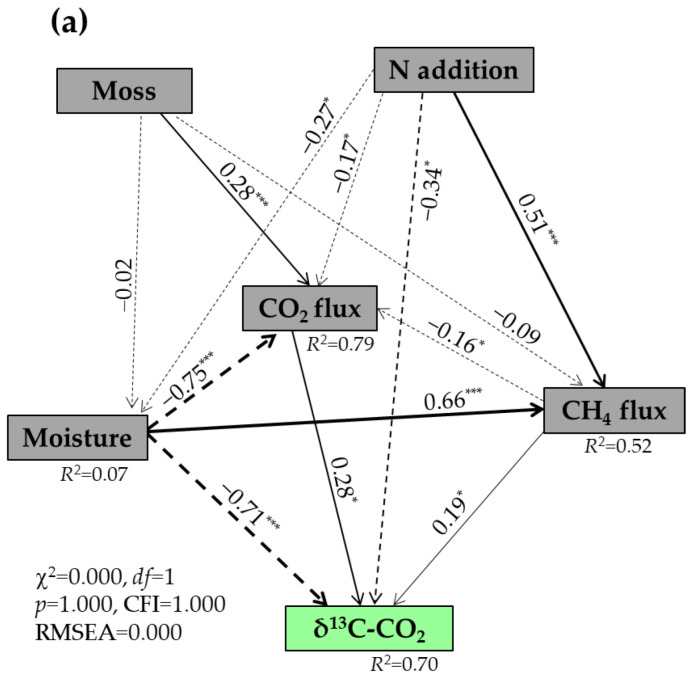
Structural equation modeling (SEM) analysis of causal relationships among N addition, moss, soil moisture, CO_2_ and CH_4_ fluxes, and δ^13^C-CO_2_ (**a**), and their standardized total effects from SEM (**b**). Single-headed arrows indicate the hypothesized direction of causation. δ^13^C-CO_2_ represents δ^13^C values of respired CO_2_. Solid and dash arrows indicate positive and negative relationships, respectively. The width of arrows is in proportion to the intensity of the relationship. The numbers near the arrows are the standardized path coefficients, and *R*^2^ values indicate the proportion of variations interpreted by relationships with other variables. χ^2^ stands for Chi-square, CFI for comparative fit index, and RMSEA for root-mean-square error of approximation. *df* and *p* show degrees of freedom and probability level, respectively. *, *p* < 0.05; ***, *p* < 0.001.

**Figure 6 plants-14-02707-f006:**
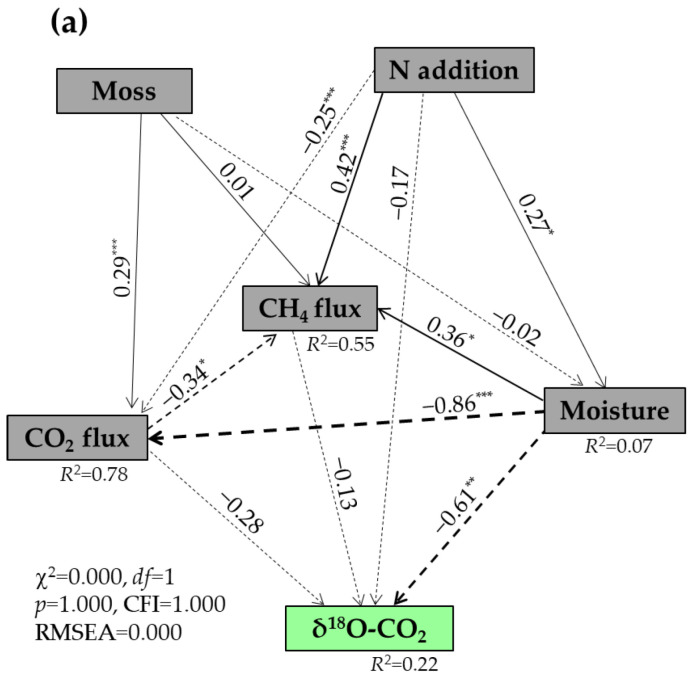
Structural equation modeling (SEM) analysis of causal relationships among N addition, moss, soil moisture, CO_2_ and CH_4_ fluxes, and δ^18^O-CO_2_ (**a**), and their standardized total effects from SEM (**b**). δ^18^O-CO_2_ represents δ^18^O values of respired CO_2_. Single-headed arrows indicate the hypothesized direction of causation. Solid and dash arrows indicate positive and negative relationships, respectively. The width of arrows is in proportion to the intensity of the relationship. The numbers near the arrows are the standardized path coefficients, and *R*^2^ values indicate the proportion of variations interpreted by relationships with other variables. χ^2^ stands for Chi-square, CFI for comparative fit index, and RMSEA for root-mean-square error of approximation. *df* and *p* show degrees of freedom and probability level, respectively. *, *p* < 0.05; **, *p* < 0.01; ***, *p* < 0.001.

**Table 1 plants-14-02707-t001:** Means and ranges of soil properties across all treatments in the plots with mosses and the results of ANOVA analysis ^a^.

Treatments	DOC (μg C g^−1^)	DTN (μg N g^−1^)	SUVA_254_ (L mg^−1^ C m^−1^)	MBC (μg C g^−1^)	MBN (μg N g^−1^)	pH (Water)	Bulk Density (g cm^−3^)
Control	90.2 (72.5–107.5)	14.8 (9.4–22.5)	1.92 (1.49–2.25)	604.1 (390.1–996.7)	140.5 (80.8–192.90)	5.84 (5.49–6.10)	0.95 (0.49–1.30)
Low N	95.1 (67.8–141.7)	15.4 (9.0–23.4)	2.22 (1.56–2.87)	527.5 (305.9–897.6)	113.7 (79.7–156.7)	5.45 (5.02–6.00)	1.01 (0.56–1.31)
High N	103.8 (80.2–139.2)	27.8 (17.1–52.9)	2.31 (1.85–3.45)	734.5 (432.3–1226.2)	168.2 (137.3–191.7)	5.17 (4.95–5.55)	0.90 (0.50–1.18)
ANOVA with treatment as a fixed factor and year as a random factor (*p* value)							
Treatment	0.265	0.015	0.003	0.120	0.066	0.004	0.012
Year	0.511	0.049	0.340	0.012	0.517	0.020	0.000
Treatment × Year	0.315	0.173	0.975	0.067	0.102	0.061	0.810

^a^ DOC, dissolved organic C; DTN, dissolved total N; SUVA_254_, the specific UV absorbance of soil K_2_SO_4_ extracts at 254 nm; MBC, microbial biomass C; MBN, microbial biomass N.

**Table 2 plants-14-02707-t002:** Means and ranges of δ^13^C and δ^18^O values of the CO_2_ respired from N-treated and non-treated experimental plots with and without mosses and the results of ANOVA analysis.

Treatments	δ^13^C Values of Respired CO_2_ (‰)	δ^18^O Values of Respired CO_2_ (‰)
With mosses		
Control	−23.95 (−29.14 to −21.51)	−13.89 (−32.27 to −2.49)
Low N	−24.76 (−28.32 to −21.49)	−12.50 (−21.32 to −5.87)
High N	−23.88 (−28.99 to −20.77)	−16.78 (−22.16 to −8.79)
Without mosses		
Control	−22.92 (−28.01 to −20.12)	−15.38 (−33.60 to −2.22)
Low N	−23.70 (−28.13 to −20.04)	−14.27 (−27.14 to −4.43)
High N	−23.28 (−27.75 to −20.31)	−17.48 (−28.86 to −8.77)
ANOVA with moss and treatment as fixed factors and year as a random factor (*p* value)		
Treatment	0.480	0.715
Moss	0.025	0.003
Year	0.002	0.331
Treatment × Moss	0.312	0.935
Treatment × Year	0.007	0.029
Moss × Year	0.330	0.997
Treatment × Moss × Year	0.933	0.367

**Table 3 plants-14-02707-t003:** Correlation between δ^13^C and δ^18^O values of respired CO_2_ versus CO_2_ and CH_4_ fluxes and soil properties ^a^.

	CO_2_ Flux	CH_4_ Flux	Moisture	Temperature	DOC	DTN	pH	Bulk Density	MBC	MBN	SUVA_254_
δ^13^C	0.65 **	−0.44 **	−0.74 **	0.76 **	−0.21	−0.39 *	−0.34 *	0.79 **	−0.61 **	0.32	0.03
δ^18^O	0.34 **	−0.34 **	−0.41 **	0.30 **	−0.25	−0.40 **	−0.09	0.41 *	−0.33	0.18	0.00

^a^ *, *p* < 0.05; **, *p* < 0.01. δ^13^C and δ^18^O represent δ^13^C and δ^18^O values of respired CO_2_, respectively. For other abbreviations, see Table 1.

## Data Availability

The data are contained within the article.

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
