# Peer review of "Effect of Mosses and Long-Term N Addition on δ13C and δ18O Values of Respired CO2 Under a Temperate Forest Floor"

_plants, 2025, doi:10.3390/plants14172707_

Round 1
Reviewer 1 Report
Comments and Suggestions for Authors
General Comments
In general, this is an interesting manuscript with lots of analyses. The way the statistics are described and interpreted is a bit confusing. For example lines 299-302 it says: “Compared with those with the removal of mosses, δ13C values of respired CO2 in the control and the low N treatment with mosses were, on average, reduced by approximately 1.03‰ and 1.06‰, which were significantly larger than that in the moss-covered plots treated at a high N dose (0.60‰) (P < 0.05)”, —but in the table 3 Treatment has no main effect and there is no significant moss × Treatment effect. So, I’m not sure what you are saying here.
Again, in lines 271-272, it says: “This indicated that on average, the presence of mosses contributed to approximately 17% and 13% to the soil CO2 and CH4 fluxes under the moss-covered forest floors at a given duration, respectively” —but according to the table the difference in CO2 flux was not significantly different.
Is it possible instead of doing multiple models and ANOVA and correlation analyses to reduce everything into one model as much as possible and then you could more clearly articulate what are the significant effects vs. what are not?
Also, I have suggested a few places where the wording is unclear, I would recommend, in addition to fixing the things I have indicated below, to also carefully proofread the final text one more time.
Specific Comments
Abstract: It is unclear how both of these statements can be correct as they seem contradictory: “The low N treatment amplified the moss-induced δ¹³C depletion, and minimized δ¹⁸O values of respired CO2”, and “No significant interactions were observed between mosses and N levels on the δ13C or δ18O values.”
Abstract does not indicate what the effect of the treatments on the fluxes was. I would encourage you to add a sentence describing these.
What is the 18O of the water? Isn’t that a pretty strong driver of the CO2 δ18O?
Line 159: suggested revision: “added N corresponded to twice the”
Line 166: suggested revision: “an equal amount of water was added to the control plots”.
Line 168: it’s not clear what “annually dominated” means?
Line 169: not clear what “certify” in this context means.
Line 213: it’s difficult to tell from the picture if the chamber is opaque or not, i.e. is this light flux or dark flux?
Line 220: I don’t think “certify” is quite the right word here.
Line 226: do you have a citation for this method?
Line 235: I think this should be “specific” rather than “special”.
Line 270: suggest replacing “varied” with “decreased”
Line 274: any thoughts on why the CH4 flux decreased when the mosses were removed?
Line 299: revise: “Compared with those with the”
Reviewer 2 Report
Comments and Suggestions for Authors
The work is very interesting. A high level of research is represented. The methodology is well described. Comments mainly concern editorial errors and discussion. The discussion repeats a lot of information from the description of the results. No clear comparison with results of other authors. References need to be rewritten.

Reviewer 3 Report
Comments and Suggestions for Authors
Please refer to the attachment.

Round 2
Reviewer 3 Report
Comments and Suggestions for Authors
Manuscript much improved. Just a small detail
Line 112: "...input, can..."
Author Response
We have made some corrections by considering the reviewer's comments. The quality of Figures and Tables was improved.